# Refractive Indices of Biomass Burning Aerosols Obtained from African Biomass Fuels Using RDG Approximation

**Emmanuel Sarpong** [1,†] , **Damon Smith** [2,‡] , **Rudra Pokhrel** [1] , **Marc N. Fiddler** [3] and **Solomon Bililign** [1,*]

1 Department of Physics, North Carolina Agricultural and Technical State University, Greensboro, NC 27411 USA; esarpong1@aggies.ncat.edu (E.S.); rppokhrel@ncat.edu (R.P.)
2 Applied Sciences and Technology Program, North Carolina A&T State University, Greensboro, NC 27411, USA; dmsmit13@aggies.ncat.edu
3 Department of Chemistry, North Carolina Agricultural and Technical State University, Greensboro, NC 27411, USA; marc.fiddler@gmail.com
* Correspondence: Bililign@ncat.edu; Tel.: +86-13362852328
† Current Address: Department of Physics and Optical Science, UNC Charlotte, NC 28223, USA; esarpon1@uncc.edu.
‡ Current Address: Department of Chemistry and Physics, Western Carolina University, Cullowhee, NC 28723, USA.

**Abstract:** Biomass burning (BB) aerosols contribute to climate forcing, but much is still unknown about the extent of this forcing, owing partially to the high level of uncertainty regarding BB aerosol optical properties. A key optical parameter is the refractive index (RI), which influences the absorbing and scattering properties of aerosols. This quantity is not measured directly, but it is obtained by fitting the measured scattering cross section and extinction cross section to a theoretical model using the RI as a fitting parameter. We used the Rayleigh–Debye–Gans (RDG) approximation to retrieve the complex RI of freshly emitted BB aerosol from two fuels (eucalyptus and olive) from Africa in the spectral range of 500–580 nm. Experimental measurements were carried out using cavity ring-down spectroscopy to measure extinction over the range of wavelengths of 500–580 nm and nephelometry to measure scattering at three wavelengths of 450, 550, and 700 nm for size-selected BB aerosol particles. The fuels were combusted in a tube furnace at a temperature of 800 °C, which is representative of the flaming stage of burning. Filter samples were collected and imaged using tunneling electron microscopy to obtain information on the morphology and size of the particles, which was used in the RDG calculations. The mean radii of the monomers were 27.8 and 31.5 nm for the eucalyptus and the olive fuels, respectively. The components of the retrieved complex RI were in the range of $1.31 \leq n \leq 1.56$ and $0.045 \leq k \leq 0.468$. The real and complex parts of the RI increase with increasing particle mobility diameter. The real part of the RI is lower, and the imaginary part is higher than what was recommended in literature for black carbon generated by propane or field measurements from fires of mixed wood samples. Fuel dependent results from controlled laboratory experiments can be used in climate modeling efforts and to constrain field measurements from biomass burning.

**Keywords:** refractive index; biomass burning aerosols; RDG theory; sub-Saharan African biomass fuels

## 1. Introduction

Among the factors that affect the climate, few are as diverse or as challenging to understand as the impacts of aerosols [1]. Consisting of minute particles suspended in the atmosphere, aerosols are emitted through a wide range of natural and anthropogenic processes and are transported around the globe by wind and weather. Unlike the long-lived greenhouse gases, aerosols have relatively short

lifetimes ranging from less than one day to a few weeks [2]. Once airborne, they affect the climate both directly, through scattering and absorption of solar radiation, and indirectly, through their impact on cloud properties. Atmospheric aerosol particles have a strong influence on global climate and the Earth's radiation budget [3]. Changes in the composition of the atmosphere can alter the Earth's climate, by changing how much infrared radiation from the surface is retained by the atmosphere, or by altering the amount of solar radiation absorbed or redirected back into space [4].

The radiative impacts of atmospheric aerosols on climate are complicated and poorly understood. These aerosol-radiation interactions are thought to have a net cooling influence on global climate, but the extent to which they do so remains uncertain [5,6]. A major uncertainty in the direct aerosol radiative forcing is the single scattering albedo (SSA), the ratio of the scattering cross section to the extinction cross section (sum of scattering and absorption) [7,8]. A relatively small error in estimating the SSA can change the sign of aerosol radiative forcing [9]. A substantial portion of this uncertainty stems from limited understanding of the spatiotemporal variability of absorption by aerosols and how this depends on particle composition and atmospheric conditions [10]. The amount of absorption and scattering of light by aerosols has been difficult to quantify, due, in part, to the challenges in measuring aerosol optical properties, which depend on the size, shape, black carbon content (and thus fuel type and combustion conditions), chemical composition, age of the aerosol, wavelength dependence of these properties, and changes in the morphology of the aerosol [11–13].

A major component of atmospheric aerosols comes from biomass burning due to wildfires or the domestic use of wood fuel. Biomass burning is one of the largest sources of organic aerosols in the atmosphere, with as much as 30% of aerosol mass belonging to black carbon (BC), which is a highly absorbing type of aerosol [10,14].

The radiative properties of biomass burning emissions from different North American, European, and Latin American fuels under different environments have been studied over the years [15–19]. However, there have only been a few studies on African biomass burning emissions [20–22], and no laboratory studies have been conducted, despite Africa representing the bulk of global biomass burning (BB) emissions [23–25]. Recent studies estimate that ~55% of the global contributions to BB aerosols come from Africa, and these emissions are expected to grow [26–29]. Anthropogenic emissions of organic carbon from the continent could be of a similar magnitude to its BB emissions by 2030 [22]. Due to the lack of data, current models being used to study air quality and climate change in Africa use data from global inventories, which are primarily collected from North America, Europe, and Asia, and are not consistent with satellite observations over Africa [22]. To our knowledge, there have not been any laboratory studies to measure the optical properties of BB aerosols using biomass fuels from Africa, making this study the first laboratory study of biomass fuels from Africa. We have been conducting laboratory measurements of optical and chemical properties of several biomass fuels from Africa. Two samples (eucalyptus and olive) are used in the modeling study described in this paper.

Modeling of scattering and absorption of light by fractal aggregates has been performed primarily by using approximate and numerical techniques [30–34]. The T-matrix method has been increasingly applied to the computation of electromagnetic scattering by multi-sphere clusters [35–44]. In this work, the Rayleigh–Debye–Gans (RDG) theory is employed [12,45], which accounts for the actual size and morphology of aggregates found in soot [46]. This method gives well-correlated results as good as those of the rigorous discrete-dipole approximation [47]. RDG is used to calculate the scattering and absorption properties of fresh and aged BB aerosol aggregates if the monomeric particles can be approximated by spheroids. High-resolution tunneling electron microscopy (TEM) images of individual carbonaceous soot particles have shown that they exist in the form of clusters of small, nearly spherical monomers (spherules), and the traditional Lorenz–Mie theory [35] may not be applicable. These spherules have diameters typically ranging from 40–60 nm [12]. The size of the monomer is described by the size parameter of the monomer in the aggregate given by $x_p = 2\pi a/\lambda$ [12], where $a$ is the radius of the monomer, and $\lambda$ is the wavelength of the incident light.

In this work, we compare experimentally measured optical properties (extinction and scattering cross sections) of BB aerosol using cavity ring-down spectroscopy (CRDS) and nephelometry to theoretical results obtained using the Rayleigh–Debye–Gans (RDG) approximation. MATLAB is used to compute the radiative properties of BB aerosol based on the Rayleigh–Debye–Gans approximation, as outlined in Section 3. By fitting the theoretical values to the experimental results, the size dependent refractive indices (RI) were obtained for BB aerosol from two fuel sources.

## 2. Experiment Method

The experimental method referred to as the "extinction-minus-scattering technique" is used to determine the optical properties of BB aerosol. This technique measures both the extinction and scattering coefficients, and the absorption coefficient is obtained as the difference between the two [19,48]. Details of the experimental approach and the error analysis are described in our earlier work [49,50]. Details of the smog chamber and the production of BB aerosol are described in [51], and we only provide a brief description here.

For laboratory samples, BB aerosols were generated by combusting each wood sample in a tube furnace. The furnace (Carbolite Gero, HST120300-120SN) holds an 85 mm outer diameter (OD), 80 mm inner diameter (ID, and 750 mm long quartz working tube, and has a heated region of 300 mm as described in Smith et al. [51]. Stainless steel mounts and insulation plugs on either end enable the introduction and sampling of gasses. The smoke and gasses produced from combustion were sent directly to the chamber via a heated (200 °C), $\frac{1}{4}$-inch OD stainless steel transfer tube. The characteristics of the smog chamber used in this work are described in Smith et al. [51]. BB particles were taken from the smog chamber via graphite impregnated silicone tubing before entering a 710 nm impactor inlet (3.8 μm diameter cut point), neutralizer, and long differential mobility analyzer (DMA, TSI, Model 3080), where the aerosol was size selected. Flow through the entire system (0.58 sL/min) was produced by a pump within the condensation particle counter, and the DMA sheath flow was 2.8 L/min in single blower mode. Aerosol flow then entered a ring-down cavity, where the aerosol extinction was measured over a range of wavelengths between 500 and 580 nm at 2 nm increments, and the extinction cross section $C_{ext}$ (μm$^2$/particle) was found using the measured extinction coefficient $\alpha_{ext}$ (1/μm) and the number density of particles $N_{CRD}$ (particles/cm$^3$) in the cavity. After the CRDS, aerosol scattering coefficients were measured at 453, 554, and 698 nm using the integrating nephelometer (TSI, model 3563), with particle concentration measured by the water-based condensation particle counter (WCPC, TSI, model 3788). The absorption cross section was derived from the difference between the extinction and scattering cross sections for each particle size considered. Particles with central mobility diameters of 200, 300, and 400 nm were selected with the DMA. We have determined that the size distributions, specifically the range of diameters passed to the system by the DMA, showed no overlap. The standard deviations for each size were; 200 ± 21, 300 ± 31, and 400 ± 42.

It has been shown that the presence of large, multiply charged particles passed by the DMA can artificially increase measured cross sections, even if their number density is relatively small [52]. Other groups have shown that measured extinction coefficients exceeded the predicted ones for 100 and 200 nm particles, which are most affected by the "multiple size–multiple charge" problem [53]. As such, only particles 200 nm or greater were considered in this work. However, even with 200 nm particles, it has been shown that a small DMA sizing error can still produce significant changes in the extinction [54]. In principle, errors in the DMA must be corrected [55,56]. However, we did not make corrections due to DMA sizing error in this work.

For this work, BB particles generated from combustion were impacted onto a 3 mm copper tunneling electron microscopy (TEM) 400 mesh grids (TED PELLA, INC, 01844) with a pore diameter of 10 μm, using a small pump at a flow rate of 1 L/min. The grid provides structural support for the sample and allows for easy and safe handling of the sample, while a carbon film provides a uniform and conductive surface. The TEM grid was placed in the mouth of a 1/8 inch Swagelock fitting and was held in place with stainless steel tubing having an outer diameter (OD) of 1/8 inch and an inner

diameter (ID) of 1/16 inch. The samples for the TEM analysis were specifically prepared to a thickness of (100–200 nm) to allow electrons to be transmitted through them. The filter samples were imaged using a Transmission Electron Microscope (TEM) (Carl Zeiss Libra 120 Plus) operating at a 120 kV accelerating voltage using lanthanum hexaboride as the electron source. The TEM at the Joint School of Nanoscience and Nanoengineering (JSNN) used for imaging in this work has a high precision stage with a ±75° tilt and contains an Omega energy filter, which allows the selection of distinct electron energies for specimen viewing. The instrument includes an Olympus Mega View G2 cooled charge-coupled device (CCD) camera for digital viewing and image capturing. The resolution of the camera is 1376 × 1032 pixels with 14-bit dynamic range. Since the wavelength of electrons is much smaller than that of light, the TEM images have better resolution than that of a light microscope. A low intensity beam and short exposure time of at most 200 ms was used to avoid beam damage to the sample during imaging.

## 3. Theoretical Method

The scattering properties of non-spherical particles can be either measured experimentally or computed theoretically [57]. Fitting the theoretical results to experimental results provided a means to extract important optical properties such as the real and imaginary parts of the refractive indices of BB aerosol.

The morphology of a dry BB aerosol as a fractal cluster is represented by a statistical scaling law [58,59] as:

$$N_p = k_o \left( \frac{R_g}{a} \right)^{D_f},$$ (1)

where $N_p$ is the number of monomer spherules in the aggregate, $R_g$ is the radius of gyration, $a$ is the radius of the monomer, $k_o$ is the fractal pre-factor (also called structure coefficient) [60], and the fractal dimension, $D_f$, describes the dimension of the monomer distribution, which has been reported to be within 1.6–1.9 for soot produced from biomass burning [12,17,60,61]. Typically, the structure of the agglomerate is characterized by the fractal dimension and the fractal pre-factor [62], and, in general, $D_f < 2$ represents a lacey aggregate while $D_f > 2$ represents a compact aggregate [45]. Equation (1) holds when $N_p$ is sufficiently large [45]. The radius of gyration is a measure of the overall radius of the aggregate, which is a parameter required by RDG approximation, and is given by:

$$R_g^2 = \frac{1}{\sum_i R_i^2} \sum R_i^2 \left( a_{i\text{-}CM}^2 + R_{g,i}^2 \right),$$ (2)

where $R_i$ is the radius of the $i$th primary particle, $a_{i\text{-}CM}$ is the separation distance of the $i$th monomer from the center of mass of the particle, and $R_{g,i}$ is the radius of gyration of the $i$th primary particle.

The RDG approximation, though an approximation, plays a vital role in the study of light scattering by aggregates [59]. The RDG approximation is relatively simple, but its application requires an in-depth understanding of its range of validity, which has been reviewed by several authors [47,63,64] and the references therein.

The validity range of the RDG approximation is based on satisfying (i) the conditions that both $2x_p|m-1| \ll 1$ and $|m-1| \ll 1$ are satisfied [47,65,66], where $m$ is the complex refractive index and $x_p$ is the optical size parameter, and (ii) the effects of multiple scattering induced by other particles in the aggregate and self-interactions of primary particle itself are negligible. The first condition is not often satisfactorily met by flame generated soot due to its relatively large refractive index [66,67], but it is also shown that RDG accurately reproduces scattering properties of soot [66]. Unlike the discrete dipole approximation, Mie theory, and T-matrix approach, the RDG approximation is computationally less intensive and less expensive, and thus lends itself to the efficient evaluation of fractal-like aggregates [68]. The RDG approximation is accurate enough to be used as a numerical tool for estimating the optical properties of particles with refractive index near unity and having any shape [48]. The detailed formalism

and derivations outlined here for the RDG approximation are based on the work by Köylü and Faeth [67] and by Liu et al. [68].

　　Application of this theory to fractal soot aggregates further assumes that (i) the primary particles in the aggregate are spherical and are of the same diameter (monodisperse), and (ii) the primary particles in the aggregate are in point-contact without any overlap. The absorption $C_a^p$ and scattering $C_s^p$ cross sections are given by [64,65]:

$$C_a^p = \frac{4\pi x_p^3}{k^2} Im \frac{m^2 - 1}{m^2 + 2}, \tag{3}$$

$$C_s^p = \frac{8\pi x_p^6}{3k^2} Im \left[ \frac{m^2 - 1}{m^2 + 2} \right]^2, \tag{4}$$

where $m$ and $k$ are the complex refractive index and wavenumber, respectively. The superscript $p$ denotes the properties for primary particles or monomers. The differential scattering cross section of an aggregate of a given size, averaged over all orientations of each aggregate within a statistically significant monodisperse aggregate population, is given by Equation (5) [63]:

$$C_{vv}^{agg}(\theta) = \frac{C_{hh}^{agg}(\theta)}{cos\theta^2} = N_p^2 C_{vv}^p S(qR_g), \tag{5}$$

where $C_{vv}^p$ is given by

$$C_{vv}^p = \frac{x_p^6}{k^2} Im \left[ \frac{m^2 - 1}{m^2 + 2} \right]^2, \tag{6}$$

and the double subscripts identify the direction of the polarization of the incident and scattered light, where *vv* represents vertical-vertical, *hh* represents horizontal-horizontal, the superscript *agg* denotes the properties for the aggregate, $q = 2k\sin(\theta/2)$ is the modulus of scattering vector, $\theta$ is the scattering angle, which is defined by directions of incident and scattered light, and $S(qR_g)$ is the structure factor. The structure factor was originally introduced and applied by Debye to the scattering of an ensemble of particles to account for the structure of the aggregate. Based on the structure factor, two regimes can be distinguished thus: in the small angle ($qR_g < 1$), called the Guinier regime, and in the large angle ($qR_g > 1$), called the power-law regime. A unified approximate expression for the structure factor for the entire range of the two regimes has been introduced by Yang and Koylu [46]. This expression is adopted in this work, which is given by Equation (7):

$$S(qR_g) = \left[ \frac{1 + 8(qR_g)^2}{(3D_f) + (qR_g)^8} \right]^{-\frac{D_f}{8}}. \tag{7}$$

　　Given the structure factor, the total scattering cross section for unpolarized incident radiation can be computed using Equation (8):

$$C_s^{agg} = N_p^2 C_{vv}^2 2\pi \int_0^\pi S(qR_g) \frac{1}{2} \left(1 + \cos^2 \theta\right) \sin \theta d\theta. \tag{8}$$

In RDG approximation, the interaction among primary particles is completely neglected as far as absorption is concerned [69]. The aggregate absorption cross section is simply the summation of all the primary particle cross sections, which is obtained by multiplying Equation (3) by the total number of primary particles $N_p$. Therefore, the absorption cross section for the aggregate is given by Equation (9):

$$C_a^{agg} = N_p C_a^p. \tag{9}$$

Based on the RDG approximation, an iterative algorithm was developed in MATLAB (see supplementary material) to compute the optical properties (scattering cross section, absorption cross section, extinction cross section, and the single scattering albedo (SSA)) that best fit the values measured in the lab. The key input parameters for the calculations are the particle's mean monomeric radius, $a$; the estimated number of primary particles, $N_p$; the wavelength of incident light, $\lambda$; the real and imaginary parts of the refractive index, $m$; the fractal dimension, $D_f$; and fractal pre-factor, $k_o$. Generally, the fractal dimension and fractal pre-factor are determined to be in the range of 1.67 to 1.83 and 2.05 to 2.90, respectively, for fresh non-collapsed BB aerosol [12]. In this work, $D_f$ and $k_o$ are taken to be 1.8 and 2.2, respectively, following [46], which have been shown to represent the morphology of flame-generated aggregates in different combustion environments.

*Image Processing and Data Extraction from TEM Image*

The morphology of BB aggregates can be characterized in a simple manner due to their fractal-like nature. The TEM images of fractal-like aggregates produced by burning biomass fuels are analyzed to extract metrics on the size, shape, and morphology of the aerosol particles. The RDG approximation treats the aerosol particles as fractal-like aggregates composed of small primary spheres. The number of these primary spheres, $N_p$, is related to the overall aggregate size, $R_g$, through Equation (1). The primary particle diameter and radius of gyration of the aggregate were determined using ImageJ software. For the RDG calculation, mean monomer size was used from several selected monomers for each sample. From the projected area of the primary particles ($A = \pi d^2/4$), the average monomer diameter is obtained by averaging the diameter of several selected distinguishable particles. The overall aggregate size, $R_g$, was determined based on the recipe for image characterization of fractal-like aggregates [70] by measuring the aggregate projected maximum length, $L$. The aggregate projected maximum length, $L$, shown in Figure 1, and the overall size of the aggregate, $R_g$, are related by Equation (10):

$$\frac{L}{2R_g} = 1.50 \pm 0.05 \,. \tag{10}$$

The quantity, $L$, is easily measurable, and, once obtained, the overall size of the aggregate can be found from Equation (10). In determining $L$ from the image, three measurements were done, and the software (ImageJ) was recalibrated each time. The average value of $L$ is what is reported here. With $k_o$, $D_f$, $a$, and $R_g$ determined, the number of monomers, $N_p$, in the aggregate is estimated through Equation (1); the results of which areratjion listed in Table 1.

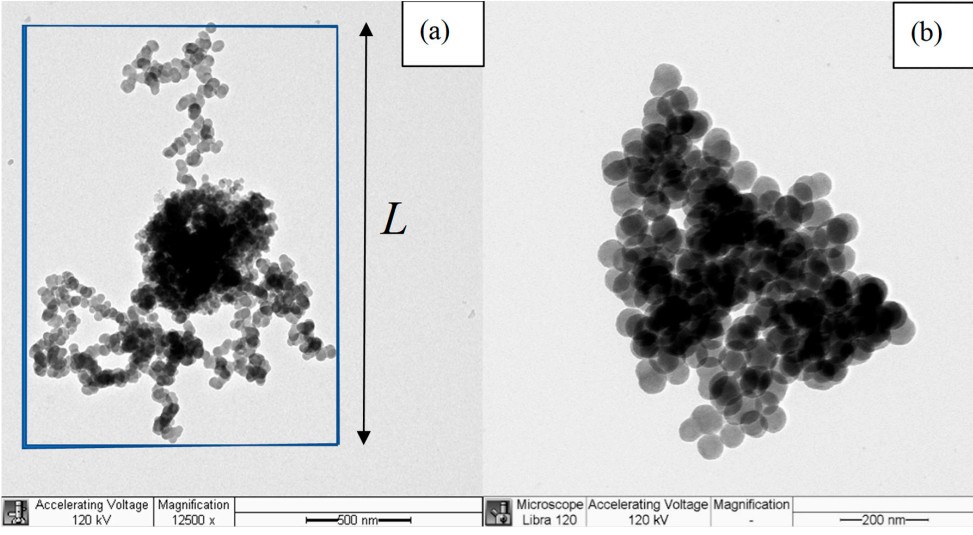

**Figure 1.** Examples of TEM images of biomass burning aerosol particles from (**a**) eucalyptus fuel and, (**b**) olive fuel.

**Table 1.** Input data from images of filter samples from eucalyptus and olive fuels.

| Fuel | No. of Particles Analyzed | $a$ (nm) | $L$ (nm) | $R_g$ (nm) | $N_p$ |
|---|---|---|---|---|---|
| Eucalyptus | 37 | 27.8 | 1972.9 | 657.6 | 654 |
| Olive | 43 | 31.5 | 902.2 | 300.7 | 128 |

## 4. Results

In this work, we treated the fresh BB aerosols as fractal aggregates, and the measured scattering and extinction coefficients were used to derive an effective refractive index from RDG approximation. This was accomplished by iteratively adjusting the real and imaginary parts of the refractive index until the calculated extinction and scattering cross sections best matched the experimentally measured ones for a given size distribution over a selected wavelength region of 500–580 nm used for the modeling. The single scattering albedo (SSA) was then derived from the ratio of the scattering cross section and to the extinction cross section and compared to the measured SSA. Common biomass fuels in East Africa, eucalyptus and olive, were selected for this modeling study based on available measurements in our lab. Optical properties (extinction, scattering, and SSA) were measured for size-selected fresh BB aerosols representing flaming combustion.

### 4.1. Eucalyptus Fuel Combusted at 800 °C

In this work, the size distribution was obtained from the TEM images of BB aerosol collected on filters following combustion. From the projected area of the primary particles, the average monomer diameter was measured to be 55.6 nm by measuring the diameter of several selected distinguishable particles. A histogram of monomeric particle radii (not diameters) is shown in Figure 2 for eucalyptus, along with a Gaussian fit that has a standard deviation of 2.9 nm. The primary particle diameters in flame-generated agglomerates have been shown to follow a normal distribution [71] as displayed in Figures 2 and 5. The TEM analysis of primary particle radius of aggregates from the combustion of eucalyptus is shown in Figure 1. The mean diameter obtained is well within 50 ± 10 nm for the scanning electron microscopy (SEM) image of the freshly emitted biomass aerosol as reported by Manfred et al. [12]. This size distribution leads to a size parameter, $x_p$, in the range of $0.30 \leq x_p \leq 0.34$ for measurements performed at wavelengths between 500 and 580 nm.

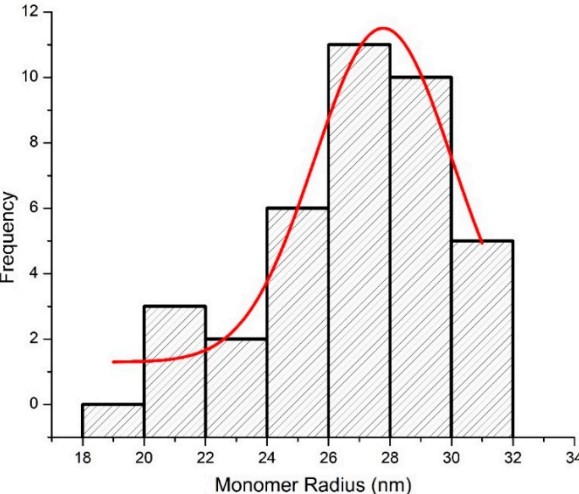

**Figure 2.** TEM analysis of monomer radius of aggregates generated from biomass burning of eucalyptus fuel.

The SSA serves as a gauge for measuring the absorption or scattering nature of aerosol particles and is the main parameter used to determine the direct radiative forcing of the particles [72]. The experimental mean SSA values of the eucalyptus BB aerosol and their standard deviations were 0.46 ± 0.02, 0.44 ± 0.01, and 0.42 ± 0.02 for particle mobility diameters of 200, 300, and 400 nm, respectively. Based on the measured fire-integrated modified combustion efficiency (MCF$_{FI}$), which is the ratio of the concentration of $CO_2$ divided by the sum of the concentrations of CO and $CO_2$, of 0.994, combustion at 800 °C is considered a flaming phase combustion. Flaming phase-dominated fires tended to produce an aerosol with high mass fractions of strongly light-absorbing elemental carbon [73,74]. This measured low SSA indicates that the particles are highly absorbing in the visible spectral range, which is consistent with the assumption of higher concentrations of BC, with estimated SSA as low as 0.2–0.3 [75,76]. We, therefore, initially assumed a larger imaginary part for the refractive index in the RDG calculations. The $n$ values were varied from 1.2 to 1.8 in steps of 0.01 and the $\kappa$ values from 0.02 to 0.8 in steps of 0.001. The refractive index was retrieved from the extinction and scattering cross section at multiple wavelengths for each of 200, 300, and 400 nm particle size. For each measured scattering and extinction cross section, the best combination of $n$ and $\kappa$ were retrieved. The real and imaginary components of the refractive index are iterated to minimize Equation (11) [77]:

$$X^2 = \sum_{i=1}^{N_\lambda} \frac{\left(C_{ext,exp} - C_{ext,calc}\right)^2}{\varepsilon_i^2} + \sum_{i=1}^{N_\lambda} \frac{\left(C_{sca,exp} - C_{sca,calc}\right)^2}{\varepsilon_i^2}, \tag{11}$$

where $\varepsilon$ is the error in the individual measurement, $N_\lambda$ is the number of wavelengths, $C_{ext,exp}$ is the experimental extinction cross section, $C_{ext,calc}$ is the calculated extinction cross section, $C_{sca,exp}$ is the experimentally measured scattering cross section, and $C_{sca,calc}$ is the calculated scattering cross section. This retrieval method is appropriate for monodispersed aerosol particles whose refractive index does not vary strongly with wavelength over the selected wavelength region [77].

The resulting fits for cross sections are presented in Figure 3a–c for mobility diameters of 200, 300, and 400 nm, respectively. Refractive indices were found to be 1.31 + i0.045, 1.48 + i0.132, and 1.56 + i0.206 for the same respective sizes, as shown in Table 2. While fits were generally good, we were unable to reproduce the observed wavelength dependence with any RI. This is likely due to a small wavelength dependence of the real portion of the RI. The mean SSA obtained from the RDG fitting to the experimental values and their standard deviations were 0.47 ± 0.01, 0.44 ± 0.01, and 0.42 ± 0.01 for the same respective sizes. These closely match the experimental averages shown in Table 2. Figure 4 shows the observed and derived SSA values from the models as a function of wavelength. While almost no wavelength dependence is observed experimentally, SSA values resulting from fit refractive indices show a slight wavelength dependence, which decreases more quickly with wavelength. This is also likely the result of a small wavelength dependence of the real portion of the RI.

**Table 2.** Mean single scattering albedo (SSA) values from experimental and Rayleigh–Debye–Gans (RDG) calculations and retrieved refractive index for eucalyptus fuel.

| Particle Size (nm) | RDG Mean SSA | Measured Mean SSA | Retrieved Refractive Index |
|---|---|---|---|
| 200 | 0.47 ± 0.01 | 0.46 ± 0.02 | 1.31 + i0.045 |
| 300 | 0.44 ± 0.01 | 0.44 ± 0.01 | 1.48 + i0.132 |
| 400 | 0.42 ± 0.01 | 0.42 ± 0.02 | 1.56 + i0.206 |

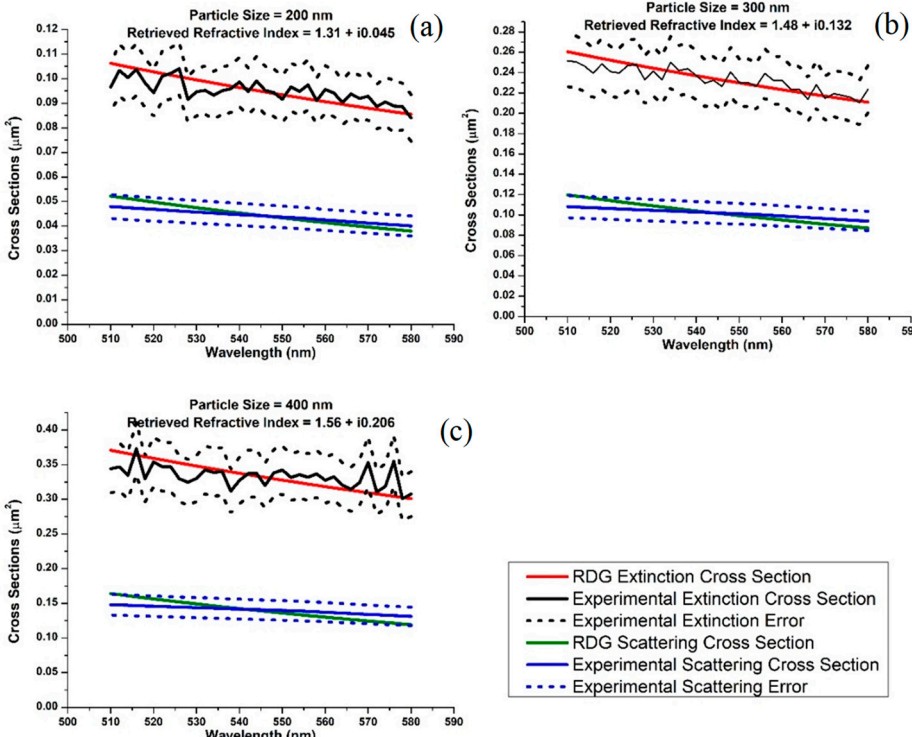

**Figure 3.** Comparison of measured and calculated cross sections for eucalyptus fuel, for (**a**) 200, (**b**) 300, and (**c**) 400 nm sized particles.

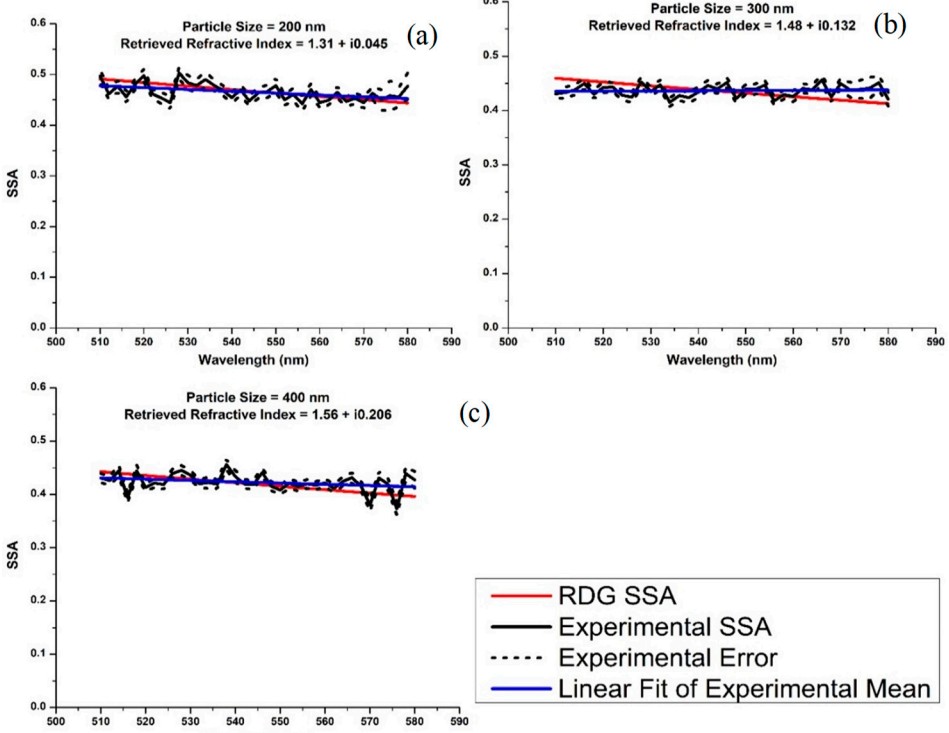

**Figure 4.** Comparison of measured and calculated SSA values for eucalyptus fuel, for (**a**) 200, (**b**) 300, and (**c**) 400 nm sized particles.

### 4.2. Olive Fuel Combusted at 800 °C

The mean diameter of the monomers obtained for olive fuel is 63 nm, a little higher than for eucalyptus fuel. A histogram of monomeric particle radii is shown in Figure 5 for olive, along with a Gaussian fit that has a standard deviation of 4.2 nm. Compared to the eucalyptus, olive monomers are somewhat more variable in their diameters. For this, a size parameter, $x_p$, is derived in the range of $0.34 \leq x_p \leq 0.39$ for wavelengths 500–580 nm. For this fuel, only two sizes of 200 and 300 nm were considered. The measured values at 400 nm were highly inconsistent and therefore not used. The $MCE_{FI}$ of olive combustion was 0.994 at 800 °C, which is indicative of flaming stage combustion.

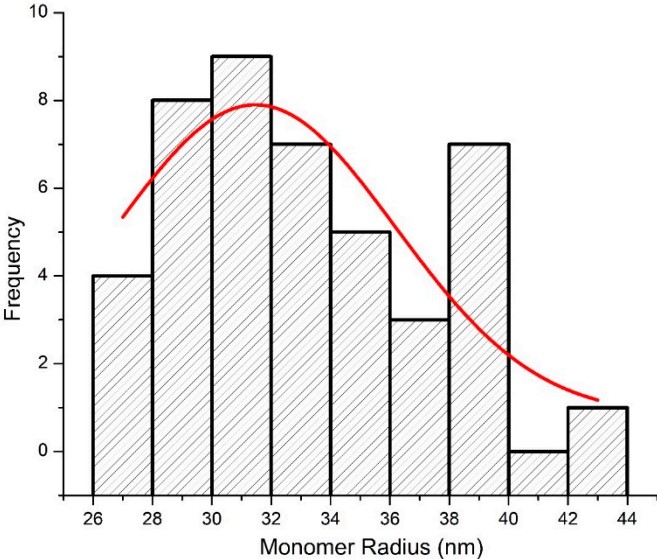

**Figure 5.** TEM analysis of monomer radius of aggregates generated from biomass burning of olive fuel.

For the 200 and 300 nm diameter particles, the respective retrieved refractive indices were 1.44 + i0.242 and 1.50 + i0.468, as shown in the fits in Figure 6a,b. Compared to the eucalyptus fuel, the real parts of the refractive index are nearly equal, but there is a significant difference in the imaginary part of the refractive index. In the case of the olive fuel, the imaginary part tends to increase sharply with increasing particle size. The high imaginary parts obtained in this work may suggest a high content of BC in the combusted fuel. Interestingly, we obtained the same mean SSA values for both the calculated and measured data of 0.28 ± 0.01 for the two particle sizes considered, as displayed in Table 3 and Figure 6c,d. In this case, the SSA was independent of the particle size. This result was somewhat different than what was observed for the eucalyptus fuel, where the mean SSA very slightly decreased with increasing size and increasing refractive index. The BB aerosol particles produced from the olive fuel have very low mean SSA values and high imaginary parts of the RI, indicating that the particles are more absorbing in the visible spectral range than those produced from eucalyptus fuel.

**Table 3.** Mean SSA values from experimental and RDG calculations and retrieved average refractive index for olive fuel.

| Particle Size (nm) | RDG Mean SSA | Measured Mean SSA | Retrieved Refractive Index |
|---|---|---|---|
| 200 | 0.28 ± 0.01 | 0.28 ± 0.01 | 1.44 + i0.242 |
| 300 | 0.28 ± 0.01 | 0.28 ± 0.01 | 1.50 + i0.468 |

Like the eucalyptus fuel, while the extinction and scattering cross sections decreased gradually with increasing wavelength (Figure 6a,b), the SSA shows a very weak wavelength dependence (Figure 6c,d). Unlike eucalyptus, the SSA values calculated from the cross sections produced from the fit refractive indices match the observed wavelength dependence.

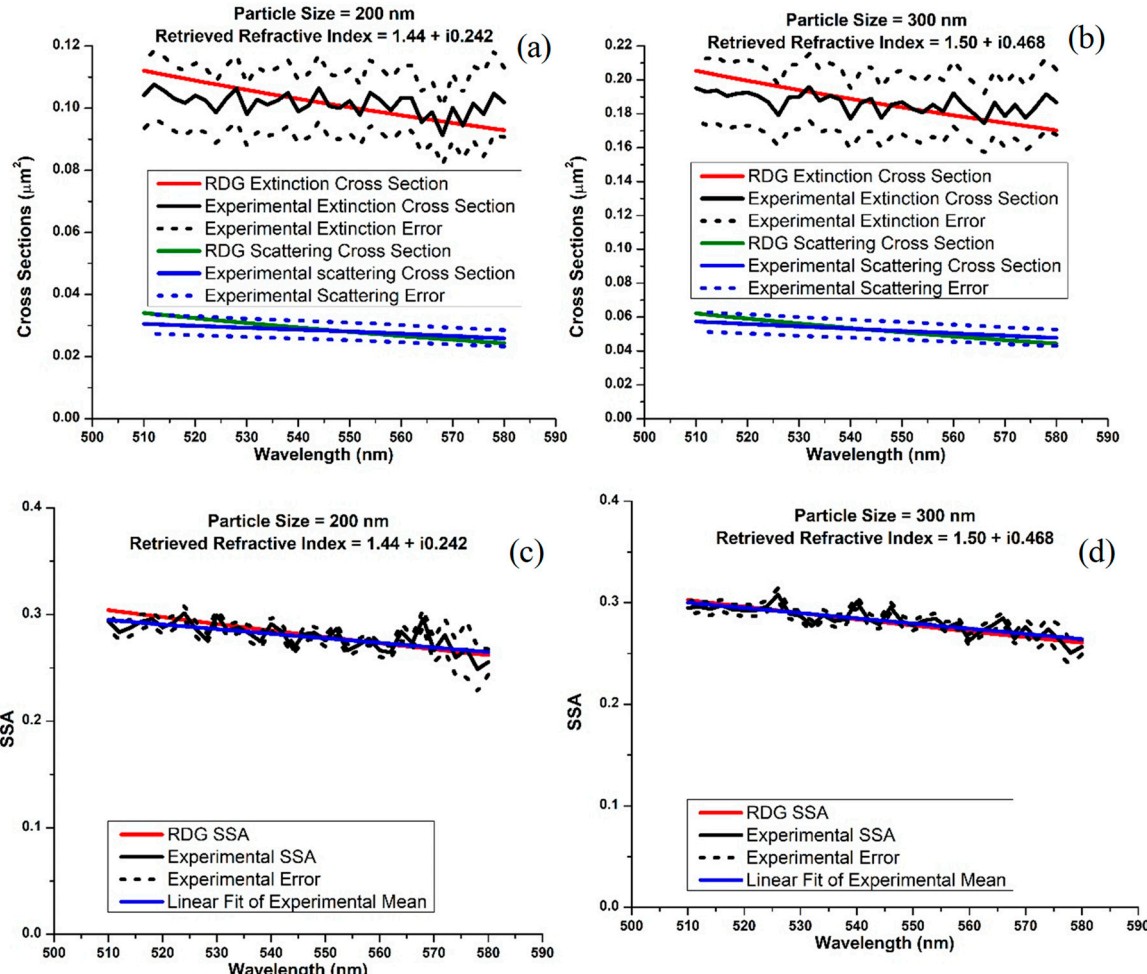

**Figure 6.** Comparison of measured and calculated cross sections and SSA values for olive fuel. Subfigures show (**a**) the cross sections for 200 nm size particles, (**b**) the cross sections for 300 nm size particles, (**c**) the SSA for 200 nm size particles, and (**d**) the SSA for 300 nm size particles.

## 5. Discussion

The optical properties of combustion aerosols strongly depend on particle size, shape, BC content, fuel type, and combustion conditions. As a result, a wide range of values of optical properties are reported in literature for BB aerosol particles. This reflects the dynamic nature of vegetation fires, variations in smoke ageing processes, differences in measurement techniques [78,79], and assumptions used in modeling these properties. Consequently, fundamental parameters such as the complex refractive index of the particles, and the percentage of major constituents, namely black (or elemental) carbon (EC) and organic carbon (OC), are highly uncertain.

The refractive indices for BB aerosols reported in literature take a range of values. This is due to the fact that the refractive index depends on the size of the aerosol, the mixing state, the wavelength at which it is estimated, and the water content of the particles [80]. In the case of aerosols from biomass burning, one of the most important factors to consider when it comes to estimating the refractive index is the EC/OC ratio, which determines the relationship between absorption/dispersion of the radiation by the aerosol. Unfortunately, those measurements were not available during this work.

It is often very difficult to link aerosol optical properties to specific fuels or combustion conditions in field measurements during wildland fires, either from the ground or from aircraft [78,79,81]. It is also difficult to measure and separate flaming and smoldering phase emissions during field measurements, since both processes are occurring simultaneously in a wildfire. Laboratory biomass burning studies are useful for quantifying biomass burning emissions because a specific quantity of fuel is burned

and most of the emissions are captured, neglecting any losses before sampling. Burning conditions, such as temperature, air flow and composition, and humidity, can be controlled, while particle size selection and tunable lasers can also be used to determine size-dependent and wavelength-dependent properties, respectively.

In both cases, for eucalyptus and olive fuels, the measured SSA values were very low. It has been reported that aerosol optical properties are strongly dependent on combustion conditions [82]. Flaming-dominated combustion produces particles mostly composed of EC, but smoldering-dominated combustion does not. The higher mass fractions of EC in an aerosol produced by flaming-dominated fires lead to particles with low SSA (in the mid-visible) and a weak wavelength dependence of absorption [83]. The SSA in these cases was as low as 0.4 [84], 0.45, 0.32, and 0.26 at wavelengths of 400, 532, and 600 nm during the Fire Laboratory at Missoula Experiments (FLAME) 4 performed at the U.S. Forest Service Fire Science Lab (FSL) in Missoula, Montana [85]. Our experimental results are consistent with these findings.

Mack et al. [86] conducted studies of North American biomass fuels based on FLAME 2 measurements. In these studies, a total of 21 chamber burns were performed using 18 fuels. Absorption was measured using a photoacoustic spectrometer at 532 nm, and scattering was measured using a nephelometer at 450, 550, and 700 nm. They measured SSA values ranging from 0.43 to 0.99, leading to a refractive index ranging from 1.54 to 1.67 for the real part and 0.011 to 0.217 for the imaginary part, retrieved using Mie Theory. These results suggest that SSA values and refractive indices varied significantly. The variations were attributed partially to fuel composition and condition (fresh, dry, etc.) and partially to the combustion conditions.

At $\lambda$ = 355 and 532 nm, Alados-Arboledas et al. [87] retrieved the complex refractive index of a freshly emitted BB plume with a particle diameter of 200 nm, and found their RI to be 1.49–1.53 and 0.02 for the real and imaginary components, respectively. Aerosol optical depth (AOD) was measured using sun-photometers and back-scatter coefficients, and Raman Lidar inversion algorithms were used to extract extinction coefficients, particle size, and complex refractive index, from which the SSA was computed with a Mie-scattering algorithm that assumed spherical particles. SSA values were higher than those in this work, ranging from 0.76 to 0.9, which is consistent with the low imaginary part of the refractive index.

Dubovik et al. [88], using eight years of worldwide distributed data from the Aerosol Robotic Network (AERONET) network of ground-based radiometers, reported the aerosol absorption and other optical properties in several key locations. The SSA values for biomass fuels from African savannah showed a decrease from 0.88 to 0.78 with increasing wavelength from 440 nm to 1020 nm. The retrievals of the real and imaginary parts of the refractive index for biomass burning smoke varied by region. The refractive index retrieved for African savannah was 1.51 for the real part and 0.021 for the imaginary part. While the real portion of the RI is commensurate with this work, their imaginary portion is significantly smaller, likely due to the presence of OC.

Using simulations to investigate the radiative forcing of biomass burning during wildfire events and AOD measurements in rural Spain, Alonso-Blanco et al. [89] estimated a refractive index of about $1.48 - 0.005i$ for a mean AOD of 0.5, and a refractive index of about $1.45 - 0.004i$ for a mean AOD of 0.4, both of which have very low imaginary components of the RI compared to values derived in this work.

From the combustion of African hardwood (musasa) under atmospheric conditions with a relative humidity (RH) less than 10%, Hungershoefer et al. [79] reported a refractive index of $1.56 - i0.01$ for fresh smoke at $\lambda$ = 550 nm using Mie calculations. For savannah grass it was $1.6 - i0.01$. Surprisingly, the imaginary part was very low, even though there was a large concentration of elemental carbon. The lower imaginary component was attributed to a positive bias in EC data due to the presence of high molecular weight organic substances.

Retrieved dry aerosol real refractive indices generally ranged from 1.56 to 1.59 from combustion measurements during the Yosemite Aerosol Characterization Study (YACS), where smoke from numerous wildfires active in the western United States was analyzed [73]. Real parts between 1.53 and 1.58 at wavelengths of 438, 670, 870, and 1020 nm were derived from BB emissions from cerrado vegetation using spectral sun/sky data measured by the AERONET radiometers in Brazil [90]. Literature values for the imaginary part of the effective refractive index show a much larger variability with values between 0.0093 and 0.1 [88]. Our real values were commensurate or slightly smaller than these reported values, while imaginary parts tended to be larger.

The results from Brazil by Yamasoe et al. [90] suggested a small effect of water vapor on the refractive index. However, they stated that measurements in the eastern United States with high humidity and high humidification factors did show a strong reduction of the refractive index with increase of the total perceptible water vapor. An important factor that can modify the role of aerosols in the global energy budget is the RH. Aerosol particles can take up water and become larger in size. Water uptake results in more than just a change in size, it also changes particle mass, composition, shape, morphology, and optical properties [91]. At ambient RH of 48–80%, Guyon, Boucher [92] reported a refractive index of $1.41 - i0.013$ at a wavelength of 545 nm in field measurements conducted in the Amazon tropical forest. This value corresponds to SSA values ranging from 0.90 to 0.93. These particles are highly scattering, and the lower imaginary refractive index component is reasonable. Our measurements were done at 0% RH under dry conditions and comparisons with measurements made at higher RH may not be appropriate.

Black carbon is assumed to have a commonly used refractive index of $1.75 - 0.630i$ [15]. In the literature, the range of refractive index for black carbon was 1.2–2.0 and 0.1–1.0 for the real and imaginary parts, respectively [93]. Our values are within these broad ranges.

For black carbon formed through pyrolysis of propane, refractive indices determined using RDG and Mie theory range from 1.92 to 2.0 for the real part and 0.63 to 0.67 for the imaginary part [94]. These values are close to the values recommended for biomass black carbon by Bond and Bergstrom [15].

For the eucalyptus fuel, the mean SSA values ranged from 0.47 to 0.42 for the RDG calculations and 0.46 to 0.42 for the measured values, with no discernible dependence on wavelength (shown in Figure 4), but the extinction and scattering cross sections show wavelength dependence (shown in Figure 3) as reported in our recent papers for pine [18,50]. The cross sections were observed to decrease with increasing wavelength. For the olive fuel, the SSA values and the cross sections show a weak wavelength dependence.

Based on the experimental values, the ranges of RIs retrieved using the RDG approximation for eucalyptus were 1.31–1.56 and 0.045–0.206 for the real and imaginary parts, respectively, while for olive they were 1.44–1.50 and 0.242–0.468, respectively. These RI correspond to flaming stage burning for fresh emissions in the visible spectrum (510–580 nm) for near zero RH. These results can be compared with measurements made under nearly similar conditions.

Our derived real part of the refractive index is within the range of most reported values, even though they were derived under different conditions; however, the derived imaginary part of the refractive index is larger than the reported values for fresh smoke. This is a result of the very low SSA measured in our experiments. The SSA values reported in the literature for fresh smoke are much larger than our results. This may be because in most field and laboratory measurements, burning conditions are different and often one cannot separate the different the forms of carbon emitted or distinguish between combustion conditions.

Our results suggest that SSA values, and hence refractive indices of BB aerosols, are strongly dependent on burning conditions, fuel type, and morphology. For fresh emissions, Mie theory is not the best approach. The particles generated at a temperature of 800 °C had a low SSA and a larger imaginary part of the RI, which implies that they consist of more elemental carbon. Additionally, they are not spherical, as Mie theory assumes.

The RDG approximation, which considers the fractal nature of the biomass particles, has been able to reproduce the measured optical properties of BB aerosols. It was shown that for black carbon fractal aggregates, there are no spherical models at any size or RI that well represents scattering of light in the visible [76]. Our effort to use Mie theory to reproduce the experimental results led to a RI of $1.6 - i0.8$ for olive, which is a much larger real and imaginary part of the refractive index. This could be attributed to errors introduced due to either multiply charged particles passed by the DMA that are not accounted for in measuring extinction and scattering cross sections or known issues with Mie theory in dealing with fractal aggregates.

Any method used to correct for the presence of multiply charged particles must be compared with an experimental method, such as using an aerosol particle mass analyzer (APM) to take an aerosol stream that has been size selected with a DMA and separate it by mass, where particles that have the same electrical mobility, but different mobility diameters could be separated. Following the recommendation of Radney and Zangmeister [95], we plan to use a second charge neutralizer after the DMA and prior to the APM to reduce and quantify the effect of multiply charged particles in future work.

## 6. Conclusions

By retrieving the best fit complex refractive index to fit measured extinction and scattering cross sections, we have modeled the optical properties of BB aerosol in the spectral range of 500–580 nm using the RDG approximations.

Two fuel samples from East Africa (eucalyptus and olive) were studied. The fuels were combusted at 800 °C, filter samples of the freshly emitted aerosols were collected under 0% RH, and were imaged using TEM. The real part of the retrieved refractive index was found to be in the range of $1.31 \leq n \leq 1.56$ and the imaginary part was in the range of $0.045 \leq k \leq 0.468$. The refractive index was sensitive to particle size, as larger particles tended to have a higher real and imaginary portions of their RI for both fuels.

From the laboratory experiments of fresh smoke samples of biomass fuels commonly burned in the United States during FLAME 2, Mack et al. [86] estimated the SSA for 18 different fuels at $\lambda = 532$ nm to be 0.428–0.990. This indicates extensive variations in the chemical composition of smoke aerosol. The SSA values obtained in this work from the African fuels were in the range of $0.28 \leq \omega \leq 0.46$.

The results presented use two fuels that may not be a general representation of the myriad of fuels burned daily in Africa, but since there are no experimental measurements of BB aerosols from African fuels, this may be the first attempt to provide accurate measurements to be used in understanding the impacts of BB aerosols on the regional climate of Africa. Future work will be conducted on a variety of African fuels under different conditions of temperature and humidity, and under the influence of dark and photochemical aging. Our retrieved refractive index of the BB aerosol will be instrumental in estimating the radiative impact of BB aerosols on regional and global climate.

**Supplementary Materials:** The MATLAB code and the experimental data are available online at http://www.mdpi.com/2073-4433/11/1/62/s1.

**Author Contributions:** S.B. and M.N.F. conceived and designed the experiments; D.S. performed the laboratory experiments; E.S. carried out the TEM imaging, performed the theoretical calculations and analyzed the data; S.B., R.P. and E.S. wrote the paper. All authors have read and agreed to the published version of the manuscript.

**Funding:** This research was funded by the US National Science Foundation, grant number NSF-AGS 1555479 and this work was performed in part at the Joint School of Nanoscience and Nanoengineering, a member of the Southeastern Nanotechnology Infrastructure Corridor (SENIC) and National Nanotechnology Coordinated Infrastructure (NNCI), which is supported by the National Science Foundation (Grant ECCS-1542174).

**Acknowledgments:** The authors wish to thank Kyle Nowlin from the Joint school of Nanoscience and Nanoengineering for his role as a trainer in using the tunneling electron microscope. We would also like to thank Samin Poudel from the Computational Science Department for his tremendous help with writing MATLAB code.

**Conflicts of Interest:** The authors declare no conflict of interest.

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
