# Peer review of "Refractive Indices of Biomass Burning Aerosols Obtained from African Biomass Fuels Using RDG Approximation"

_atmosphere, doi:10.3390/atmos11010062_

Round 1
Reviewer 1 Report
The manuscript "Refractive Indices of Biomass Burning Aerosols Obtained from African Biomass Fuels using RDG Approximation" by E.Sarpong et al., describes a method to obtain the refractive indices of biomass burning (BB) aerosols from two sources, eucalyptus and olive. The authors made experimental measurements and used the Rayleigh-Debye-Gans (RDG) approximation to retrieve the refractive indices in the spectral range 500-580 nm. I congratulate the authors for an excellent job.
The first part of the manuscript presents the experimental procedure and then the use of the RDG approximation is explained. The applicability to two of these sources of BB is examined in detail.
The paper is adequate for the journal scope, it is well written, well structured and the subject is a useful contribution, so it deserves publication.
I have just a few specific comments:
- references 10 and 15 are the same;
- authors may consider the inclusion of a nonbreakable space between numbers and %, as well as, between numbers and units and between Figure/Table and the respective numbers;
- references 70 and 71 have the misplaced accents;
- pg.5, line192: misplaced accents;
- equation (5): missing symbol;
- equation (10);
- figures 2 and 5 can be smaller;
- pg.8, line 297: missing symbols.
Reviewer 2 Report
This is a valuable work but in my opinion it needs considerable revision. Unfortunately, I have problems with the structure of the manuscript too.
My biggest concern is that although the manuscript suggests that these are multi-wavelength measurements but actually they are not really. Therefore, I suggest to rearrange the manuscript in a way that it primary focuses on the experimental and theoretical results at 550 nm, where measurement results for both scattering and extinction are available. The wavelength dependency of the refractive index should be examined separately and the limitations stem from the lack of sufficiently wide wavelength range of extinction measurements has to be clearly stated. I also suggest to mention the concept of Angström exponent in this wavelength dependency part of the manuscript.
There are several problems with SSA too. To start with in line 53 it is written: “the fraction of radiation the particle scatters rather than absorbs”. This is misleading as SSA is the scattering divided by extinction not by absorption. Please give the correct definition of SSA! Furthermore, the manuscript is also very misleading concerning the role of SSA in the retrieval of the refractive index several times. Already in the abstract (line 20) it is written: “This quantity is not measured directly, but it is obtained by fitting the measured scattering cross section, extinction cross section, and the single scattering albedo…”. However, it can be seen from Equation 11 that only the scattering and the extinction cross sections are used to retrieve the refractive index and not the SSA. There are several other places in the manuscript where SSA should not be there. Although SSA is very important, it should not be mixed up with the refractive index. Rather SSA should be discussed separately.
I also suggest to discuss the consequences of the lack of direct (e.g. photoacoustic) measurement of the absorption. Would it improve the reliability of the measurements as far as the refractive index and SSA is concerned?
There are other minor problems with the manuscript:
Complex numbers are treated in a very confusing way. Line 32: “The complex refractive index increases with increasing particle mobility diameter.” This sentence does not make sense: a complex number does not increase. The amplitude of a complex number can increase. Or it is possible that both the real and the imaginary part of the complex number increase. Furthermore, a complex number does not have real and complex part (line 33 and repeatedly later in the text several times) rather it has real and imaginary part (as it I written correctly in line 160).
Line: 275 refers to Figure 3 as a TEM analysis while actually Figure 3 is about optically measured cross sections.
There is no definition (or a reference to a definition) of “Modified combustion efficiency” in the manuscript.
Author Response
see responses attached
